# CGT: CLUSTERED GRAPH TRANSFORMER FOR URBAN SPATIO-TEMPORAL PREDICTION

## ABSTRACT

Deep learning based approaches have been widely used in various urban spatio-temporal forecasting problems, but most of them fail to account for the unsmoothness issue of urban data in their architecture design, which significantly deteriorates their prediction performance. The aim of this paper is to develop a novel clustered graph transformer framework that integrates both graph attention network and transformer under an encoder-decoder architecture to address such unsmoothness issue. Specifically, we propose two novel structural components to refine the architectures of those existing deep learning models. In spatial domain, we propose a gradient-based clustering method to distribute different feature extractors to regions in different contexts. In temporal domain, we propose to use multi-view position encoding to address the periodicity and closeness of urban time series data. Experiments on real datasets obtained from a ride-hailing business show that our method can achieve 10%-25% improvement than many state-of-the-art baselines.

## 1 INTRODUCTION

The aim of this paper is to use urban data to study spatio-temporal prediction problems, whose goal is to forecast region-based spatial distribution in the future (Shi & Yeung, 2018; Wang et al., 2019). Recently, region-based spatio-temporal forecasting has been extensively studied with various applications in traffic (Li et al., 2018b), ride-hailing services (Zhang et al., 2018; Li & Zheng, 2019), environment (Liang et al., 2018), resources (Li & Zheng, 2019), and human flows (Wang et al., 2019; Shi et al., 2020), among others. Such spatio-temporal forecasting is crucial for various tasks, such as dispatching and pricing, in urban computing (Zheng et al., 2014). Accurate spatiotemporal forecasting methods may not only lower the barrier for decision making, but also improve the quality of various services including traffic management, transportation, and air quality management. However, achieving accurate prediction represents major challenges for many existing methods due to some difficulties in urban data, such as unsmoothness.

As an illustration, we will show throughout the paper that urban spatio-temporal prediction task suffers from not appropriately handing spatial and temporal unsmoothness. Unsmoothness is a common phenomenon in many spatial and/or temporal data sets. The observations in some locations/time-steps differ substantially from the observations in their neighboring locations/time-steps. For example, rush-hour traffic surge is temporally unsmoothed, while it occurs in many regions in a city. Figure 1 shows potential spatial unsmoothness due to different points of interest. Specifically, $S_1$, $S_2$ and $S_3$ denote business, residential, and recreational areas, respectively, whereas $U_1$, $U_2$, and $U_3$ are their boundaries. Traffic observations in the interior areas of $S_1$, $S_2$ and $S_3$ may be quite close to each other, whereas observations along $U_1$, $U_2$, and $U_3$ may have much larger variations. Figure 2 shows 24-hour temporal patterns in the spatial neighborhood of a selected smooth region (2a) and a selected unsmooth region (2b). The temporal patterns for the smooth region are

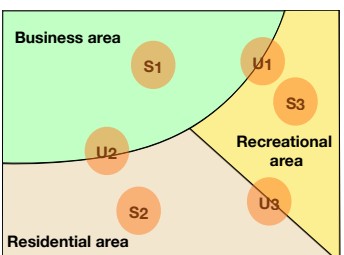

Figure 1: An example of spatial distribution for illustrating spatial smoothness and unsmoothness, where $S_1$, $S_2$ and $S_3$ are smooth function areas and $U_1$, $U_2$ and $U_3$ are on their boundaries.

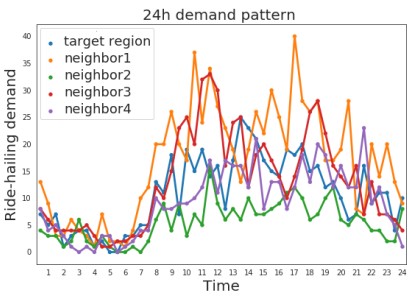
(a) The close neighborhood of a
smooth region (blue line)

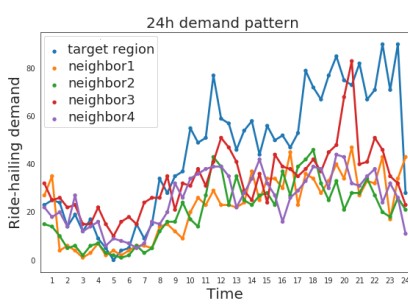
(b) The close neighborhood of an
unsmooth region (blue line)

Figure 2: The 24-hour demand patterns of selected smooth and unsmooth regions (in blue lines) and their close neighbors (in other colors). In Figure 2a, the target region and neighbor 2-4 are smooth regions, while the neighbor 1 is an unsmooth region. In Figure 2b, the target region is an unsmooth region. As shown in an interpretation in Section 4.3, the spatial learner assigns neighbor 1 in Figure 2a and target region in Figure 2b to the same cluster. Remaining regions in Figure 2a are assigned to another cluster.

close to those of most of their closest neighbors except the region with yellow color [1] in terms of shape and magnitude. The temporal patterns of the unsmooth region differ more significantly from those in all his closest neighbors. Moreover, we also discover some sharp surge and drop patterns in the unsmooth region. For more quantitative metrics for measuring unsmoothness, please refer to Appendix A for details.

However, the attempt to handling unsmoothness is quite limited both at model and feature levels. Many existing prediction methods share their spatial and temporal feature extractors universally across all time-steps and/or all regions. Thus, it greatly limits the expressiveness of those prediction methods, since they are greatly vulnerable to the presence of unsmoothness in urban data.

In this paper, we develop a novel Clustered Graph Transformer (CGT) framework by integrating Graph Attention Network (GAT) (Veličković et al., 2018) with Transformer (Vaswani et al., 2017) based on a unified attention encoder-decoder architecture (Bahdanau et al., 2015). The GAT and Transformer are effective methods for spatial and temporal feature extraction based on attention mechanism. To handle spatial unsmoothness, we develop a novel Clustered Graph Attention Network (CGAT) by leveraging different graph attention kernels on different regions. The CGAT learns clustering assignments for each region according to their temporal patterns so that regions with different temporal patterns are assigned with different spatial feature extractors. To handle temporal unsmoothness, we propose additivity-preserved multi-view position encoding (MVPE) by characterizing different kinds of temporal relationship including weekly or daily periodicity and temporal closeness (Zhang et al., 2017). In summary, our major contributions are summarized as follows:

- To the best of our knowledge, CGT is the first of its kind in using graph attention network with a transformer under an encoder-decoder architecture for long sequence spatiotemporal forecasting.

- We propose to improve the expressiveness of spatial feature extraction using the clustering technique. We also handle the temporal unsmoothness by using MVPE, which is a tailored position encoding for urban computing.

- Compared with most existing baseline models, CGT achieves improvement ranging from 10% to 25% in spatiotemporal prediction on various tasks, datasets and cities. Error reduction on unsmooth areas dominates this improvement.

---

[1]Except region 1. Details are in section 4.3

## 2 METHODOLOGY

We formalize the learning problem of urban spatiotemporal forecasting and introduce the CGT framework to capture the spatial and temporal dependencies and unsmoothness of urban data.

### 2.1 URBAN SPATIOTEMPORAL FORECASTING

We consider urban spatiotemporal forecasting problems in a city. Suppose that the city can be divided into $V$ disjoint regions and we observe temporal information across $T$ time intervals within each region. Our spatio-temporal data is represented as a $|V| \times T$ matrix. As an example, we consider taxi demand forecasting problem and each entry in the matrix represents the total number of taxi orders inside a region within a specific time interval. Our prediction problem is formulated as $f : \mathbb{R}^{|V| \times T_x} \to \mathbb{R}^{|V| \times T_y}$ such that we use the historical data within time period $T_x$ to predict the future trend in time period $T_y$, that is,

$$\mathbf{X}_{t-T_x+1:t} = [X^{t-T_x+1}, \dots, X^t] \xrightarrow{f(\cdot)} \mathbf{X}_{t+1:t+T_y} = [X^{t+1}, \dots, X^{t+T_y}]. \tag{1}$$

Problem (1) is a *one-step prediction problem* as $T_y = 1$ and a *multi-step prediction problem* as $T_y > 1$. We regard the whole city as a graph with $V$ regions (or vertices), each of which has both demand and supply numbers across $T$ time intervals.

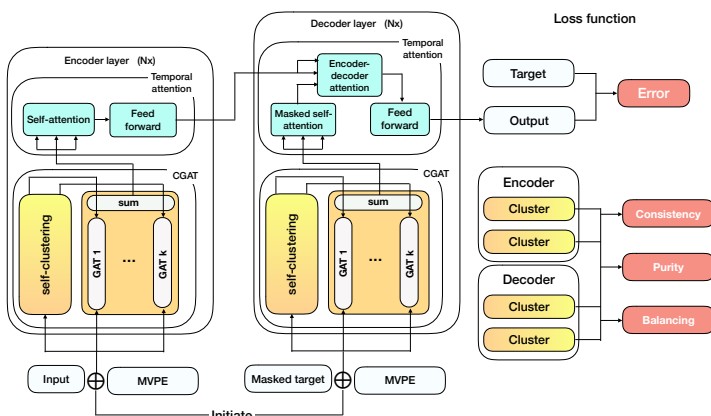

Figure 3: The CGT framework overview. CGT is an encoder-decoder architecture. The encoder and decoder consists of stacked CGATs and temporal attentions. The encoder and decoder are linked with attention. The loss function contains the prediction error, as well as three regularizers to ensure the efectiveness of clustering.

### 2.2 FRAMEWORK OVERVIEW

Figure 3 shows the overview of the CGT framework. The CGT is an encoder-decoder based architecture. The clustered graph attention layers (CGAT) and temporal attention layers are stacked alternately in the encoder and decoder. For prediction problem (1), the input tensor to encoder is $\mathbf{X}_{t-T_x+1:t}$. In the training phase, the whole target sequence $\mathbf{X}_{t:t+T_y}$ is input to the decoder and the masked attention controls the visibility of target sequence in each step of decoding. The $\mathbf{X}_t$ is used to initialize the decoder state at the initial decoding step. In the testing phase, at step $T_s$ ($T_s \leq T_y$), the decoder input tensor is $[\mathbf{X}_t, \hat{\mathbf{X}}_{t+1:t+T_s}]$, where $\hat{\mathbf{X}}$ represents predicted values from previous steps. The multi-view position encoding (MVPE) is added to input tensors in both encoder and decoder.

Our key novel contribution in CGT lies in the design and use of CGAT and MVPE. On spatial mode, CGAT integrates self-clustering and a set of graph attention kernels (GAT), which are used for region clustering and spatial feature extraction, respectively. CGATs firstly generate soft clustering assignment scores for regions according to their temporal pattern in the sampled period. The cluster scores are used as the weights to apply different GATs to each region. On temporal mode,

we use multi-view position encoding (MVPE) to manually inject ordering information, periodicity information, and closeness information to the model.

## 2.3 MULTI-VIEW POSITION ENCODING

We design a multi-view position encoding mechanism (MVPE) for urban computing. The position encoding consists of five channels as follows. There are two channels for weekly periodicity given by

$$PE_1(x) = (1 + \sin(2\pi x/7))/2 \quad \text{and} \quad PE_2(x) = (1 + \cos(2\pi x/7))/2.$$

There are two channels for daily periodicity given by

$$PE_3(x) = (1 + \sin(2\pi x))/2 \quad \text{and} \quad PE_4(x) = (1 + \cos(2\pi x))/2.$$

There is one channel for closeness given by

$$PE_5(x) = \exp(-ax^2),$$

where $x$ represents the temporal difference between a fixed timestep (e.g. the time $t+1$ in formulation (1)) and a specific timestep in inputs or outputs.

The MVPE mechanism injects ordering information and relevance information to the temporal modeling. First, MVPE provides unique identifiers for all time-steps in temporal sequences, which is critical for non-recurrent to model sequences. Second, according to Zhang et al. (2016), the weekly and daily periodicity, as well as closeness are considered to be highly important in modeling urban time series. The periodicity channels provide constant crest/trough values for timesteps spanned by an integer number of periods. The closeness channel provides a surging value for the nearby timesteps. The model will be trained to make predictions according to different temporal relevance to avoid making over-smoothing ones.

## 2.4 CLUSTERED GRAPH ATTENTION

We propose the clustered graph attention layer (CGAT) for spatial feature extraction. It learns different graph attention kernels for different regions based on a gradient-based self-clustering assignment such that different regions are treated differently in spatial dependency modeling.

First, a vertex-level soft-assignment to $K$ clusters is learnt from the temporal pattern of each vertex:

$$C = \sigma_s(\sigma_r(X_f \mathbf{W}_f)_t \mathbf{W}_t), \tag{2}$$

where $C$ is the cluster assignment score for each vertex to $K$ clusters. $\mathbf{W}_f$ and $\mathbf{W}_t$ are parameters for linear layers on the feature mode and temporal mode, respectively, and $\sigma_r$ and $\sigma_s$ represent the relu and softmax activation functions. The feature dimension of input tensor $X_f$ is first squeezed to 1 using $\mathbf{W}_f$, in order to provide summarized temporal pattern at each vertex. The $\mathbf{W}_t$ is further applied to the temporal pattern to calculate a $K-$dimensional cluster assignment score.

Second, we use $c_{x_i,k}$ to denote the assignment score of $C$ for assigning vertex $x_i$ to cluster $k$. Then, $C$ is used to re-weight the latent output by using $K$ GATs as follows:

$$h_{x_i} = \sum_{k=0}^{K} \mathcal{G}(x_i) c_{x_i,k}, \tag{3}$$

where $h_{x_i}$ is the summed output for a CGAT layer at vertex $x_i$ and $\mathcal{G}_{\theta_k}(x_i)$ is the output for each GAT at vertex $x_i$. We use the GAT defined by Veličković et al. (2018). The attention between two vertex $x_i$ and $x_j$ is calculated as:

$$\alpha_{i,j} = \frac{\exp(\sigma(\mathbf{a}^T[\mathbf{W}x_i||\mathbf{W}x_j]))}{\sum_{x_m \in \mathcal{N}_{x_i}} \exp(\sigma(\mathbf{a}^T[\mathbf{W}x_i||\mathbf{W}x_m]))}, \tag{4}$$

where $\mathbf{W}$ and $\mathbf{a}^T$ are network parameters, $\mathcal{N}_{x_i}$ represents the neighborhood of region $x_i$, $[\cdot||\cdot]$ is the concatenation operation, and $\sigma(\cdot)$ is the Leakyrelu activation function. The aggregation function for GAT is given by $\mathcal{G}(x_i) = \sigma(\sum_{x_j \in \mathcal{N}_i} \alpha_{i,j} \mathbf{W}x_j)$.

## 2.5 TEMPORAL ATTENTION AND TRANSFORMER

The temporal features are extracted by adapting the multi-head attention model to the temporal mode of spatio-temporal tensor (Vaswani et al., 2017). Detailed information is included in Appendix C.

## 2.6 OPTIMIZATION OBJECTIVE

The optimization objective of CGT consists of two major components. The first component is the prediction error given by

$$L_{pred} = ||\mathbf{y} - \hat{\mathbf{y}}||_2, \tag{5}$$

where $|| \cdot ||_2$ the standard $L_2$ norm and $\hat{\mathbf{y}}$ and $\mathbf{y}$ are, respectively, the tensor of predicted values and that of true values.

The second component consists of three constraints for obtaining a good vertex cluster schema $C$. Let the clustering result $C = (C_{b,l,v,k})$ be a tensor of order $(B, L, V, K)$, where $B$ is the batch size, $L$ is the number of all CGAT layers, $V$ is the number of vertices, and $K$ is the cluster number. Thus, $C_{b_1,l_1,\cdot,\cdot}$ denotes a $V \times K$ matrix and $C_{b,l,v,\cdot}$ is a $K \times 1$ vector. The three constraints including consistency, purity, and diversity are summarized below.

- Consistency: The clustering assignment is generated through the temporal pattern of each vertex such that $C$ should be temporally (batch) invariant and layer invariant. The loss function for ensuring consistency is defined as

$$L_c = \frac{4}{BL(B-1)(L-1)V^2} \sum_{b_1=1}^{B-1} \sum_{b2=b1+1}^{B} \sum_{l_1=1}^{L-1} \sum_{l_2=l_1+1}^{L} ||C_{b_1,l_1,\cdot,\cdot} - C_{b_2,l_2,\cdot,\cdot}||_2^2. \tag{6}$$

- Purity: Ideally, each vertex should be only assigned to one cluster. The loss function for encouraging purity is given by

$$L_p = \frac{1}{BLV} \sum_{b=1}^{B} \sum_{l=1}^{L} \sum_{v=1}^{V} (1 - ||C_{b,l,v,\cdot}||_2^2). \tag{7}$$

- Diversity: To balance the cluster size, we define the loss function for ensuring diversity as follows:

$$L_d = \frac{1}{BLV^2} \sum_{b=1}^{B} \sum_{l=1}^{L} ||\sum_{v=1}^{V} C_{b,l,v,\cdot}||_2^2. \tag{8}$$

The overall minimization goal is to minimize the following object function

$$\mathcal{L} = L_{pred} + \alpha(L_c + L_p + L_d), \tag{9}$$

where $\alpha$ is a tuning parameter. It is worth noting that each batch consists of training snippets within a continuous short period of time (e.g., several hours). Keeping the clusters stable by using above regularizations allows to capture the short-term stability of urban time series data, while it does not violate the daily or long-term variability. We verify this short-term stability feature in Appendix B.

## 2.7 APPLICATION-LEVEL ADAPTATIONS

Several state-of-the-art techniques need to be considered for improving our implementation and evaluation on real-world applications. First, we construct a set of graphs $\mathbb{G} = \{G1, G2, G3\}$ as the graph adjacency matrix, each of which represents region-wise distance, region-wise functional similarity and region-wise road connectivity, respectively. It follows the idea of Geng et al. (2019) in order to encode multiple region-wise relationships as multi-graphs for the better utilization of auxiliary data. Second, following Child et al. (2019), we implement spatial atrous attention to reduce the extra large spatial dimension in order to reduce the computational cost for CGATs.

Denoting atrous stride and atrous offset as $s$ and $a$, the atrous version of eq.(4) is revised to be:

$$\alpha_{i,j,g,s,a} = \frac{\exp(\sigma(\mathbf{a}_g^T[\mathbf{W}_g x_i || \mathbf{W}_g x_j])) I_{s,a}(j)}{\sum_{x_m \in \mathcal{N}_{x_i}} \exp(\sigma(\mathbf{a}_g^T[\mathbf{W}_g x_i || \mathbf{W}_g x_m])) I_{s,a}(m)}, \tag{10}$$

where $\mathbf{a}_g^T$ and $\mathbf{W}_g$ denote the feature transformation operation for graph $g$. $I_{s,a}(\cdot)$ is a masking function for atrous attention:

$$I_{s,a}(j) = \begin{cases} 1 & \text{if } j \bmod s = a; \\ 0 & \text{otherwise.} \end{cases}$$

To cover all the regions, atrous-GAT needs to traverse $a = 0, \cdots, s-1$ by in-layer aggregation. The aggregated atrous-GAT output under single graph $g \in \mathbb{G}$ is calculated as:

$$\mathcal{G}_g(x_i) = \sigma(\sum_{a=0}^{s-1} \sum_{x_j \in \mathcal{N}_i} \alpha_{i,j,g,s,a} \mathbf{W} x_j). \tag{11}$$

Each GAT outputs an aggregated output over multi-graph given by $\mathcal{G}(x_i) = \sum_{g \in \mathbb{G}} \mathcal{G}_g(x_i)/|\mathbb{G}|$. The regularizers in eq.(6)-(8) are calculated separately for multi-graph and finally averaged for calculating the loss.

## 3 RELATED WORK

### 3.1 SPATIOTEMPORAL DEEP LEARNING

There have been various ways to model the spatial and temporal dependencies with deep neural networks. The combination of CNN and RNN is the most intuitive one (Yao et al., 2018; Zhang et al., 2017; Wang et al., 2017; Liao et al., 2018; Rodrigues et al., 2019). However, CNN-based architectures are unable to handle non-Euclidean region-wise relationships. The STMGCN (Geng et al., 2019), DCRNN (Li et al., 2018b) and GCNN-DDGF (Lin et al., 2018) use ChebNet (Kipf & Welling, 2017) and diffusion convolution to extract spatial features based on rather complex graph representations. The graph attention network (Veličković et al., 2018) is the third generation of spatial modeling, which has more complex structures for discovering region-wise relationships. The ST-MetaNet (Pan et al., 2019) is a representative work of GAT with application in urban computing. In temporal feature extraction, attention-based methods outperform RNNs in terms of both accuracy and efficiency. The ASTGCN (Guo et al., 2019) is an attention-based model on both spatial and temporal domains. The GeoMAN (Liang et al., 2018) further adapts the attention-based model to encoder-decoder architecture for long sequence prediction. However, all existing methods neglect the unsmoothness problem of urban data.

### 3.2 GRAPH NEURAL NETWORKS

The research works on graph neural networks could be summarized in several categories (Zhou et al., 2018). Spectral graph convolution (Hammond et al., 2011; Kipf & Welling, 2017) defines convolution operator by the eigen-decomposition of the graph laplacian, based on the graph spectral theorem. Non-spectral graph convolution (Duvenaud et al., 2015; Atwood & Towsley, 2016; Hamilton et al., 2017) directly operates the spatial domain by defining a set of neighborhood aggregation and updating rules on vertex-level. Moreover, the gated GNN (Li et al., 2015) leverages the gating technique from LSTM, which provides more complex operations for neighborhood aggregation. GAT (Veličković et al., 2018) is using attention mechanism to discover the vertex-wise similarities in a graph. In urban computing, the intuition is confident that regions are strongly correlated. The GAT might help to discover this intrinsic structure in the region-based spatial distribution.

### 3.3 VERTEX CLUSTERING AND GRAPH POOLING

Vertex clustering is a frequently used method for graph pooling (Ying et al., 2018), which is a common technique migrated from computer vision. However, the intuition for graph pooling and vertex clustering proposed in this work is different. Graph pooling is to reduce the number of vertices such that vertices in each cluster are regarded sampled into one in the next layer. Downscaling the graph layer would improve model generality and reduce computational cost. The vertex clustering in our approach is to discover a set of vertices sharing similar patterns, which is used for better feature extraction.

### 3.4 Handling unsmoothness in machine learning

Non-local mean methods are proposed for image denoising and restoration by investigating the pixel-level self-similarity in images. Buades et al. (2005) proposed to calculate pixel pair-wise similarity within a large search window. The densities for noisy pixels are interpolated by those similar pixels. Wang et al. (2018b),Liu et al. (2018) designed non-local neural networks to incorporate the similar idea in deep learning. However, these non-local mean methods are applied to fix part of the spatial distribution at corrupted areas, which is different from our problem formulation. In time series prediction, Ding et al. (2018) proposed a extreme-value loss (EVL) for extreme event prediction in time series data. Compared with our method, EVL has no model-level effort to extract unsmooth features from data, which totally relies on the success of optimization algorithm.

## 4 Experiments

We conduct experiments on a demand/supply dataset for ride-hailing services in two cities A and B [2] from Jun 2017 to Jun 2019. The demand data is the number of ride-hailing orders in each region within each time interval. The supply data is the summation of online duration for all ride-hailing drivers. The spatial and temporal distribution is sampled and aggregated into 1km×1km map gridding and 30 minutes temporal sampling interval[3]. For both datasets, we cut off the long tail on the 95th quantile and normalize them by min-max normalization. The duration of the dataset is two years, where first 70% is used for training, 10% for validation, and remaining 20% for test. The multi-graph $\mathbb{G}$ involves region-wise relationships including neighborhood, functional similarity, and road connectivity, which are identical to those used in (Geng et al., 2019). The functional similarity graph is binarized on 0.9 so that three graphs have almost equal graph sparsity[4].

### 4.1 Overall performance

In this section, the encoder and decoder both stack 3 CABs and TABs alternately, with 16 hidden states each. The default clustering number is 3, with atrous stride being equal to 2. The number of heads in TAB multi-head attention is set to be 3. The network is optimized by using scheduled Adam on PyTorch (Paszke et al., 2017).

We conduct both one-step prediction and multi-step prediction experiments. To avoid the long sequence problem, which may exhaust computational resource, we sampled 10 input time-steps for each output time-step $T_k$. The input time-steps include 3 closest time-steps ($T_{k-1}, T_{k-2}, T_{k-3}$), 2 from daily periodicity ($T_{k-48}, T_{k-96}$) and 1 from weekly periodicity ($T_{k-48*7}$). For multi-step prediction, the default output length is 8, the encoder input length is 27.

| Data | One-step prediction | | | | Multi-step prediction |
|---|---|---|---|---|---|
| | Demand | | Supply | | Demand |
| City | City A | City B | City A | City B | City A |
| HA | 12.20 / 0.261 | 12.35 / 0.298 | 4.14 / 0.184 | 2.68 / 0.167 | 18.07 / 0.345 |
| LR | 10.61 / 0.247 | 10.08 / 0.288 | 3.22 / 0.148 | 2.12 / 0.143 | 14.31 / 0.286 |
| XGBoost | 10.28 / 0.233 | 10.25 / 0.268 | 3.12 / 0.146 | 2.07 / 0.140 | 13.03 / 0.240 |
| m-DMVST | 9.85 / 0.223 | 9.45 / 0.259 | 2.88 / 0.152 | 1.94 / 0.137 | 12.22 / 0.270 |
| m-DCRNN | 9.59 / 0.217 | 9.34 / 0.249 | 2.88 / 0.149 | 1.97 / 0.148 | 10.74 / 0.245 |
| ST-MGCN | 9.55 / 0.226 | 9.34 / 0.259 | 2.85 / 0.136 | 1.91 / 0.139 | 12.29 / 0.276 |
| m-GeoMAN | 9.97 / 0.232 | 9.86 / 0.291 | 2.94 / 0.145 | 2.00 / 0.138 | 11.29 / 0.259 |
| m-ASTGCN | 10.23 / 0.273 | 9.48 / 0.252 | 3.14 / 0.181 | 1.98 / 0.140 | 12.66 / 0.262 |
| DST-GCNN | 9.72 / 0.226 | - | - | - | - |
| m-GAT+RNN | 9.54 / 0.224 | - | - | - | - |
| Our Method | 7.29 / 0.209 | 7.66 / 0.232 | 2.66 / 0.131 | 1.73 / 0.124 | 7.49 / 0.185 |

Table 1: The error metrics (RMSE/MAPE) based on the demand-supply datasets in two cities.

---

[2]Anonymous for blind review.
[3]Designed according to industrial practices
[4]In city A, it is roughly 12000 edges for 1296 vertices.

Baselines are built based on following research works. For the sake of information fairness, all deep learning models are slightly revised to take in multiple graphs.

- HA, LR and XGBoost: Historical average, linear regression and XGBoost. The model parameters for LR and XGBoost are shared across space and time. XGBoost is implemented using LightGBM (Ke et al., 2017).
- DMVST (Yao et al., 2018): a CNN-RNN based model. Region-wise relationships are encoded as external inputs.
- DCRNN (Li et al., 2018b) and ST-MGCN (Geng et al., 2019): Both are GCN-RNN based model for spatiotemporal feature extraction. DCRNN is based on encoder-decoder architecture for multi-step traffic prediction.
- GeoMAN (Liang et al., 2018): The spatial and temporal features are extracted by spatial attention and RNN. It is based on the encoder-decoder architecture.
- ASTGCN (Guo et al., 2019): an attention-based model on spatial and temporal domain. Spatiotemporal features are extracted using GCN and convolution. ASTGCN don't have position encoding.
- DST-GCNN [5](Wang et al., 2018a): A two stream framework for traffic prediction. The first stream constructs dynamic affinity graph. The second stream use the graph and spatiotemporal convolution for feature extraction.
- GAT+RNN [5](Xu & Li, 2019): a GAT+RNN structure for traffic forecasting.

Table 1 summarizes the RMSEs and MAPEs of CGT and state-of-the-art baselines. There are few observations from above experiment results. First, the proposed CGT model outperforms all baselines in all prediction tasks by a large margin. Second, all encoder-decoder based models, including m-DCRNN, m-GeoMAN and CGT, perform well in all multi-step prediction tasks.

## 4.2 ABLATION STUDY AND PARAMETER TEST

In this section, we show the experiment results of CGT on the one-step demand prediction problem in city A with varying model configurations to show the parameter effect. As shown by figure 4, the optimal result is achieved when $\alpha = 0.1$ and $k = 3$. Table 2 shows the model performance when specific component is removed from CGT. The ablation study shows the significance of our novel contribution.

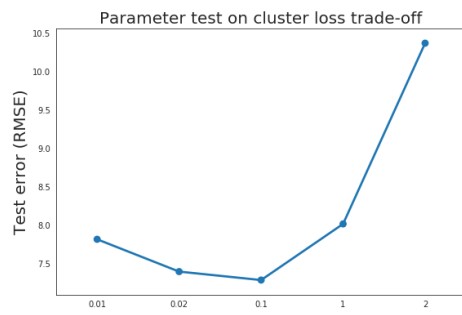 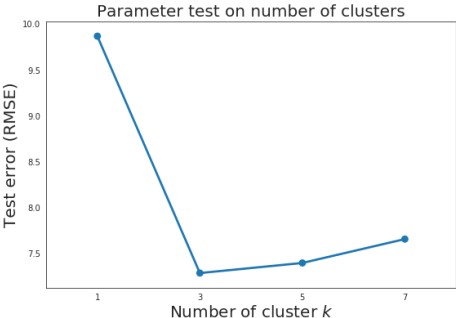

Figure 4: Parameter test for the trade-off parameter $\alpha$ and number of clusters $k$.

| Ablation component | MVPE | Clustering | MVPE and Clustering |
|---|---|---|---|
| RMSE | 8.69 | 9.87 | 10.11 |

Table 2: Ablation study for novel components

---

[5]Made-up experiment according to reviewer's comments. Unfinished due to time limit.

| Spatial | Our model | Baseline model | Error reduction | Temporal | Our model | Baseline model | Error reduction |
|---|---|---|---|---|---|---|---|
| 0-50% (smooth) | 2.64 | 3.77 | 30% | 0-50% (smooth) | 3.54 | 4.68 | 24% |
| 50-75% | 6.29 | 9.30 | 32% | 50-75% | 7.08 | 9.98 | 29% |
| 75-100% (unsmooth) | 11.51 | 17.65 | 35% | 75-100% (unsmooth) | 10.98 | 16.83 | 35% |

Table 3: Error reduction ratio on different quantiles of unsmoothness in spatial and temporal domains.

### 4.3 MODEL INTERPRETATION

We claim that the prediction error is reduced due to more accurate predictions in spatial and temporal unsmooth areas. To validate this, we explore the relationship between the unsmoothness and the error reduction from a baseline model. The baseline model is designed by removing clustering and MVPE from CGT. Figure 5 shows a positive relationship between unsmoothness and error reduction. Table 3 shows the error reduction from the baseline model on different quantiles of spatial and temporal unsmooth areas. Conclusions could be drawn that CGT could improve the prediction performance, especially in spatial and temporal unsmooth areas by significant margins.

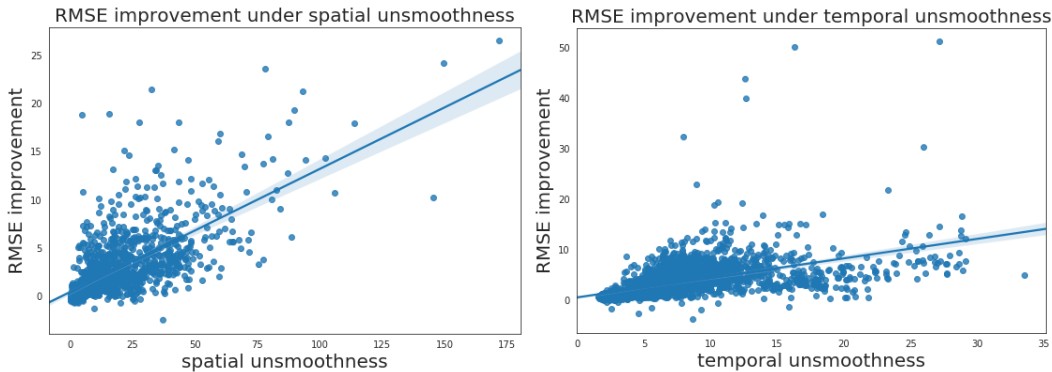

Figure 5: Error reduction and unsmoothness in spatial and temporal domains.

Above improvements on spatial mode may be caused by the effective clustering technique, which applies different spatial kernels to regions with different patterns. Referring back to Figure 2, the neighbor 1 (yellow line) in Figure 2a is an unsmooth point in the neighborhood. It is assigned to the same cluster with the target region (blue line) in Figure 2b, which is also an unsmooth region. The remaining regions in Figure 2a stay in the same cluster. According to above case study, the temporal pattern for inner-cluster regions are similar, while the temporal pattern for intra-cluster regions are diverse. It is sensible to apply a shared operator to target regions in a single cluster, since the relationship between target regions and their neighbors are more stable within a cluster. However, applying one smoothing operator to different clusters does not work very well, since it may introduce some artifacts. In conclusion, discriminating smooth and unsmooth patterns may greatly improve the local spatial feature extraction.

## 5 CONCLUSION AND FUTURE WORK

In this paper, we proposed Clustered Graph-Transformer for spatiotemporal prediction problems in urban computing. To handle the spatial and temporal unsmoothness problem, we use a gradient-based clustering technique in GAT to construct spatial feature extractor. We use MVPE and attention to handle temporal unsmoothness by providing a urban-specific temporal encoding to the temporal mode. The experiment has shown the effectiveness of the proposed model. In the future, we plan to explore long seq2seq problem using atrous transformer (Child et al., 2019) as well as explore the explanation for the attention and clustering assignments.

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

## A  PRELIMINARY ANALYSIS

In this section, we conduct a preliminary error analysis for a baseline GNN+RNN based model on spatial and temporal unsmooth areas in order to prove our assumptions and motivate our design methodology.

For input spatial temporal signal $X$, denote $X_s$ and $X_t$ as its spatial and temporal modes, respectively, and $L$ as the graph laplacian matrix for spatial distribution. The $\mathcal{L}$ is the loss metric. The spatial and temporal unsmoothnesses could be represented by the difference between a convoluted value and its true value. We define the spatial convolution kernel as $\alpha L$, which is a linear combination of the features from all neighboring locations. The temporal convolution kernel is defined as $W_t *$, which is a 1d convolution along temporal mode. The spatial and temporal unsmoothnesses are calculated as $\phi_s = \mathcal{L}(X, \alpha L X_s)$ and $\phi_t = \mathcal{L}(X, W_t * X_t)$, respectively. The $\alpha$ and $W_t$ are optimized using the same training set as the baseline model. Spatial and temporal areas with larger unsmoothness means that they are hard to be represented by using a linear combination of observations in neighboring locations.

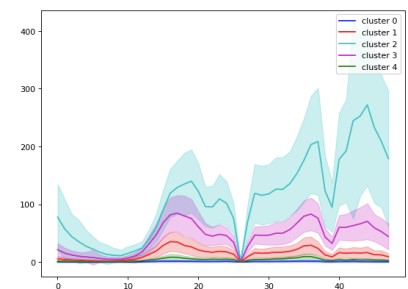

Figure 6: The averaged 24 hour temporal pattern for regions in each cluster, shaded within 1 standard deviation.

(a) Positive relationship between spatial unsmoothness (X) and prediction error (Y) for the baseline model.

(b) Positive relationship between temporal unsmoothness (X) and prediction error (Y) for the baseline model.

Figure 7: Baseline prediction error and unsmoothness

Figure 7 shows the positive relationship between baseline prediction error and spatial/temporal unsmoothness. There are several reasons for this. First, the expressiveness of GNNs are limited. Using

ChebNet as an example, it treats all neighbors in one convolution kernel homogeneously and shares its weighted transformation on feature dimension to all neighboring vertices. Thus, ChebNet lacks discriminativity within the neighborhood. Li et al. (2018a) also proves that the usage of laplacian smoothing will cause serious problem in deep-stacked GCN models. Second, spatial feature extractors are shared among all regions, that is, it is assumed that spatial feature extraction rule is universally applied to the whole city. In Figure 6, we clustered the regions in the city into 5 clusters according to their similarities by using DTW (Senin, 2008) of temporal pattern in a short snippet and print averaged 24 hour temporal pattern for them. The clustering result shows that the magnitudes for regions from different clusters are different with high confidence. The overall temporal patterns for different clusters are quite diverse.

## B  LOCAL STABILITY ON THE TEMPORAL MODE

In this section, we prove in details that the temporal patterns are stable within each batch, compared with its large variation across the whole dataset. The average value for batch-level standard deviation on temporal mode is 658.57. The standard-deviation on temporal mode for the whole dataset is 1074.84, which is significantly larger. The batch-size is the same with our experiment setting.

## C  CALCULATION FOR TEMPORAL ATTENTION IN TRANSFORMER

The temporal attention is calculated as follows:

$$Attn(Q, K, V) = softmax(\frac{QK^T}{\sqrt{d_k}})V,$$

where $QK^T$ is a self-attention to represent relationships among different time-steps. Such relationship is further applied on $V$ by transforming its temporal mode.

The feed forward layer is a linear transformation on the feature mode as follows:

$$H_{ffn}(X) = \sigma_r(X_f W_f).$$

In encoder, the self-attention is used to discover temporal dependencies in the latent feature vector $h$ such that the attention is defined as

$$Q = hW_q, K = hW_k, \text{ and } V = hW_v. \tag{12}$$

In decoder, the self-attention performs similarly, but a mask is applied on the input sequence so that the feature extraction at time $t$ could not view the temporal features later than it. The mask is implemented by an upper triangular matrix. The encoder-decoder attention transforms the temporal mode of decoder latent feature based on temporal dependencies from the encoder as follows:

$$Q = h_{enc}W_q, K = h_{enc}W_k, \text{ and } V = hW_v, \tag{13}$$

where $h$ is the decoder latent feature vector from previous decoder layers and $h_{enc}$ is the encoder feature vector which denotes the encoder state. In this application, the initial state for the decoder input sequence (target sequence) is copied from the last time-step of the input sequence.

## D  VISUALIZATION FOR PERFORMANCE ON TEMPORAL UNSMOOTHNESS

Figure 8 shows the comparison between ground truth and machine learning models based on 24-hour ride-hailing demand values in selected regions. According to the visualization, CGT model is more close to ground truth, especially in those temporal unsmoothed areas.

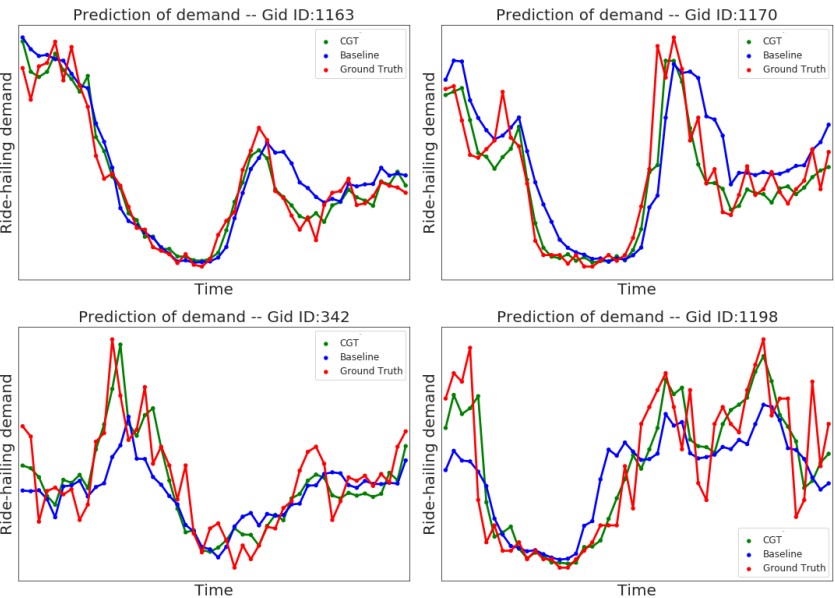

Figure 8: Plot 24-hour region demand values to compare ground truth with prediction result of CGT and baseline

