# OpenReview forum: "CGT: Clustered Graph Transformer for Urban Spatio-temporal Prediction"
_ICLR.cc/2020/Conference — Reject_

### Official Review · AnonReviewer2 · 2019-10-18
**Official Blind Review #2**

**Rating:** 3

**Review:**

Summary: This paper proposes a clustering attention-based approach to handle the problem of unsmoothness while modeling spatio-temporal data, which may be divided into several regions with unsmooth boundaries. With the help of a graph attention mechanism between vertices (which correspond to different regions), the CGT model is able to model the (originally unsmooth) cross-region interactions just like how Transformers are applied in NLP tasks (where words are discrete). Experiments seem to suggest a big improvement when compared to baselines.

Pros:
+This should be one of the first works that apply a graph transformer alike method in this domain, and specifically on the unsmoothness problem.
+ Since the dataset is not publically available, there aren't many prior works to compare the CGT to. However, at least compared to the one prior work [1] that the authors point to in Section 4, the RMSE results achieved CGT does seem to be significantly better.

========================================

However, I still have some questions/concerns on the paper, detailed below.

1) The current organization of the paper, as well as its clarity, can (and should) be significantly improved. I didn't completely understand the approach on my first two passes, and I **had** to read the code published by the authors. Here are some issues that I found:

  - For one, Figure 2 is not quite helpful as it's too messy with font size too small. A similar problem is with Figure 4 which, without further clarification (e.g., of what "atrous aggregation" exactly mean), is very hard to interpret.

  - The notations are very inconsistent and messy:
      i) In Eq. (1), you should use a symbol different from $\mathbf{X}$ to refer to the "predictions". Since you are applying $f(\cdot)$ on $\mathbf{X}_{t-T_x+1:t}$, you should not get the "exact same" target sequence. That's your target. Maybe use $\hat{\mathbf{y}}$, which you used in Eq. (8).

      ii) In Figure 3, what is the orange line? In addition, I only saw two blue lines in the figure, but the legend seems to suggest there are four of them...

      iii) The notations used in Figure 4 are somewhat confusing. For example, what does "f->1" mean? (I later found through Eq. (2) that it means transform to 1 dimension; but the small plots in Figure 4 suggest f is a "magnitude" of the feature.) In addition, there are two $H_1$ in Figure 4 with clearly different definitions.

      iv) The authors used $\mathcal{G}_{\theta_k}(x_i)$ in Eq. (3) without defining it. The definition actually came much later in the text in Eq. (6). I suggest moving the usage of the clustering assignment (i.e., Eq. (3)) to after Eq. (6).

      v) What does $[\cdot || \cdot]$ mean (cf. Eq. (4))? (The code seems to suggest it's concatenation?)

      vi) The authors first used $h_{x_i}$ in Eq. (3) to denote the output of the CAB module. Then letter $h$ is then re-used in Eq. (4) and (5) with completely different meanings. For instance, the $W_kh_i$ in Eq. (4) correspond to line 48 of the code "model.py". (By the way, nowhere around Eq. (4) did the authors explain how $h_i$ is produced, such as taking the mean over the batch dimension, etc.).

      vii) In Section 2.6, you denote the "optimal vertex cluster scheme" with letter $C$, which is used in Eq. (2). Similar for parameter $a_k$ and atrous offset $a$.

  - This not a very big problem (as it seems somewhat inevitable), but I think there are too many acronyms in the paper.

  I think it'd be great if the authors can take care of these issues, as clarity in math and descriptions are critical to the presentation of such an involuted method. It would also be useful to clearly define the dimensionality of all the variables (e.g., you defined $V$ in Section 2.1, but never used it again in later subsections).

2) Regarding the usage of the multi-view position encoding, the authors claimed that it "provides unique identifiers for all time-steps in temporal sequences". However, if you consider $x=7$ and $x=14$, then $PE_i(7)=PE_i(14)$ for all $i=1,2,3,4$ with $PE_5(7) \approx PE_5(14)$. Doesn't this invalidate the authors' claim? Also, doesn't this mean that the proposed MVPE only works on sequences with length <= 7? (In comparison, the design of positional encoding in the original Transformer doesn't have this problem.)

(You didn't show how you implemented and initialized the position encoding in the uploaded code, so I may be missing some assumptions here.)

3) In line 48 of the code (https://github.com/CGT-ICLR2020/CGT-ICLR2020/blob/master/model.py#L48), why did you take the mean over the batch dimension? Shouldn't different samples in a minibatch be very different? Does a (potentially completely independent) sample in a batch affect another sample? A similar problem occurs for Eq. (9): Why do you require clusterings of two different samples $b_1, b_2$ to be similar? (Where these samples can come from quite different times and years of the data?)

4) In the experiments, you "sampled 10 input time-steps" due to computational resources. Typically, in Transformer-based NLP tasks the sequence lengths can be over 500, with much higher dimensionality (e.g., 512); but you are only using sequence length 10 and dimensions <= 16 (in your code, you used "self.dec_io_list = [[5,8,16,16],[16,16,16,16],[16,8,8,8]]"). What is the bottleneck for the computation of your approach? (I noticed there are more than 1K vertices in city A, which may be a costly factor indeed.) How much memory/compute does the CGT method consume? How does using a longer sequence affect the performance of CGT?

5) You performed an ablation study on MVPE. Did you simply remove MVPE, or did you use the conventional PE from the original Transformer paper (Vaswani et al. 2017)? (If the latter, I'm very surprised that MVPE is so much better than PE. In that case, you may want to try MVPE on NLP tasks to see if it also improves SOTA.)

6) How did you measure unsmoothness in Figure 6? It doesn't seem like a quantifiable property to me. You should discuss this in the experiment section.

-----------------------------------

Minor questions/issues that did not affect the score:

7) There are some strange phrases/sentences in the paper. For example, the first sentence of the 2nd paragraph of Section 1: "we will show **throughout the paper** that urban spatiotemporal prediction task suffers from..."

8) Why use an encoder-decoder architecture at all? Why can't we train the model like in language modeling tasks, where we want to predict the next token? In other words, you can simply use a decoder-side CGT, and mask the temporal self-attention as in the Transformers.

-----------------------------------

In general, I think this paper proposed a valuable approach that seems to work very well on the spatio-temporal dataset they used (which unfortunately is private). However, as I pointed out above, I still have numerous issues with the paper's organization and clarity, as well as some doubts over the methodology and the experiment. I'm happy to consider adjusting my score if the authors can resolve my concerns satisfactorily.


[1] http://www-scf.usc.edu/~yaguang/papers/aaai19_multi_graph_convolution.pdf



**Experience Assessment:**

I have read many papers in this area.

**Review Assessment: Checking Correctness Of Derivations And Theory:**

N/A

**Review Assessment: Checking Correctness Of Experiments:**

I carefully checked the experiments.

**Review Assessment: Thoroughness In Paper Reading:**

I read the paper at least twice and used my best judgement in assessing the paper.

---

> ### Author Response · Authors · 2019-11-15
> **We appreciate many insightful comments from this reviewer.**
>
> 1) and 7) on paper’s organization and clarity
> -We have reorganized the technical part by following your helpful suggestions.
>
> 2) and 5).  "MVPE is not providing unique identifiers. MVPE only works on sequences with length <= 7"
> -The MVPE is providing unique identifiers to different timesteps. Considering about your example, the encodings are similar if two timesteps are considered highly-correlated. MVPE is defined on the whole temporal domain without any limitation to the sequence length. We have improved section 2.3 for better illustration.
>
> "Did you simply remove MVPE, or did you use the conventional PE from the original Transformer paper (Vaswani et al. 2017)?"
> - We simply removed MVPE from the transformer in ablation study.
>
> 3) and 4). "why did you take the mean over the batch dimension?"
> - This is a trick to reduce the computational cost due to large data dimension (1k vertices * 10 temporal slices * feature dimension). It can be considered as a pooling operator, which is similar to the “descriptor” in SENet [1].
>
> "Shouldn't different samples in a minibatch be very different. Why do you require clusterings of two different samples to be similar? (Where these samples can come from quite different times and years of the data?) "
> - We agree with your comments, but the variation of different samples within each batch is quite small for our datasets. Specifically, each batch in our design represents a short period of time. Under this batching mechanism, the spatiotemporal patterns of different samples in each minibatch are similar to each other (Please refer to appendix B for statistics). Similarly, it is reasonable to fix the cluster assignments for regions within each batch in eq(9) (eq(7) in the latest submission) and the cluster assignments are free to be different among different batches. We have added above explanation in Section 2.6.
>
> 6). "How did you measure unsmoothness in Figure 6? "
> - We use the prediction error of smoothing operators as the measurement for unsmoothness. -Related analysis and calculation for unsmoothness could be found in the appendix A.
>
> 8). "Why use an encoder-decoder architecture at all?  "
> - We use it to handle the unfixed-length prediction problem.

---

> > ### Comment · AnonReviewer2 · 2019-11-15
> > **Further comments on the authors' response**
> >
> > Thank you for your responses. The updated version of the paper is indeed better in terms of clarity.
> >
> > Given that the authors' response came in relatively late, I'm not sure how much time the authors will have to address my "further comments". But I think they could be useful anyway, should the paper be either accepted (to prepare for camera-ready version) or rejected (to improve for any likely re-submissions).
> >
> > 1) The design of MVPE still eludes me. I think the authors are proposing a very narrow and subjective design, in terms of what is "highly-correlated". Given the cycle of 7, I'm guessing that the authors want to imply there is high-correlation between the same day of different weeks. But this is subjective--- two adjacent days are also "highly-correlated". Tuesday and Thursday are "highly-correlated". (Just examples, but hopefully my point is clear here). However, the positional embeddings between $PE(1)$ and $PE(2)$ will be very different, whereas the difference between $PE(1)$ and $PE(8)$ will be minimal (actually, only slightly different in channel 5).
> >
> > 2) For the ablation study, I'm not surprised that it worsened after you removed the MVPE--- as then your model has no information about positions whatsoever. A better way of showing the usefulness of your proposed module (MVPE) is, as with any model, to compare to a baseline. In this case, e.g., the original pE by Vaswani et al.
> >
> > 3) Regarding the "mean over the batch dimension". Ok--- so if the motivation is only computational cost, what would happen if you don't do this "batch averaging"? Collapsing features along the batch dimension could result in loss of lots of information, and I'm still not sure if it's justified. If you give it enough computation time and do not take the mean over batch, I wonder if the performance is better or worse.
> >
> > 4) In your reply to the other discussant, "John Berkeley", you explained the reason for not running on city B as "cannot make up this experiment during limited rebuttal time". Have you never run multi-step prediction city B, while researching this methodology? Or maybe have you tried to start running it after the paper submission deadline? If it's available, I think it'd be useful to add, to either this version or the future versions of the paper, for completeness. If you don't have the results, I suggest you start to run it now.
> >
> > Of course, a major (and seems inevitable in this case) drawback here is that many of the tricks the authors have used are pertinent to the specific dataset they use, which unfortunately is not public. (E.g., you said  "the variation of different samples within each batch is quite small for our datasets", which is certainly not true in most of the other datasets).
> >
> > But anyway, thanks to the authors for addressing my other suggestions/problems.

---

### Official Review · AnonReviewer3 · 2019-10-23
**Official Blind Review #3**

**Rating:** 3

**Review:**

In this paper, the authors developed a neural network architecture to address the spacial and temporal unsmoothness problem, which was claimed to be neglected by existing works. The proposed model, CGT, has an encoder-decoder structure, and is characterized by clustering modules for spacial regions based on their temporal patterns. To handle temporal unsmoothness, additivity-preserved multi-view position encoding was proposed to characterize different temporal relationships. The experimental results on real ride-hailing datasets demonstrate the effectiveness of the proposed method to some extent.

The major concern is the presentation of this paper. There are many unclear points by going through the current paper, which prevents full judgement of the merits of the proposed method. First, the key problem to address is claimed to be the spacial and temporal unsmoothness. From the introduction, it is hard to see how important the problem is. It is better to use real data statistics to show the prevalence and concrete examples of such unsmoothness. Second, the technical sections presents the methods with few intuition on how does each component solve the unsmoothness problem. Figures such as fig 2 and 4 are very dense with few annotations, neither in captions nor main texts, thus are hard to understand. Finally, in the experiments, it is good to see some results for model interpretation. However, it is not clear on how to measure the spatial and temporal unsmoothness in fig 6 on the x axis.


**Experience Assessment:**

I have read many papers in this area.

**Review Assessment: Checking Correctness Of Derivations And Theory:**

N/A

**Review Assessment: Checking Correctness Of Experiments:**

I carefully checked the experiments.

**Review Assessment: Thoroughness In Paper Reading:**

I read the paper at least twice and used my best judgement in assessing the paper.

---

> ### Author Response · Authors · 2019-11-15
> **We appreciate many insightful comments from this reviewer.**
>
> 1.	"It is better to use real data statistics to show the prevalence and concrete examples of such unsmoothness."
> – We have added Figure 5 to show some concrete examples of smooth areas and unsmooth areas. Please also refer to Section 1 para. 3 for illustration.
>
> 2.	"the methods with few intuition on how does each component solve the unsmoothness problem."
>  –We have rewritten section 2.3 in order to elaborate more on the intuition of temporal unsmoothness. The intuition for spatial unsmoothness is elaborated in Sections 1 and 4.3.
>
> 3.	"Finally, in the experiments, it is good to see some results for model interpretation."
> - We have added model interpretation in Section 4.3. In Figure 4, we show the error reduction majorly comes from unsmoothness areas. In Figure 5, we show the effectiveness of the clusters, by comparing inner-cluster regions and intra-cluster regions.
>
> 4. "However, it is not clear on how to measure the spatial and temporal unsmoothness in fig 6 on the x axis. "
> We are using the prediction error of smoothing operators to measure the unsmoothness. Please see appendix A.
>
> 5. "Figures such as fig 2 and 4 are very dense with few annotations, neither in captions nor main texts, thus are hard to understand."
>  - We have improved all figures for better interpretation.

---

### Official Review · AnonReviewer1 · 2019-10-29
**Official Blind Review #1**

**Rating:** 1

**Review:**

This paper proposes to address the problem of spatio-temporal forecasting in urban data, in a way that can accommodate regions with highly distinct characteristics.

On the spatial side, they make use of Graph Attention Networks (GAT), a very recent technique for spatial feature extraction using graph attention as a form of reweighting. The authors modify the GAT to accommodate a masking that allows for selection. For some parameter K, a collection of K GATs is then combined with the masking used so that only one of them can be active at any given time. This architecture (called MGAAT) then encourages a form of clustering,
with each cluster associated with a single GAT. This modification one of the two essential contributions of the paper.
Although the description is not easy to follow, it does appear to have the potential to encourage clustering as claimed by the authors.

On the temporal side, the autoencoder-based Transformer architecture of Vaswani et al is imposed on top of the MGAAT architecture. Very few details are given in the main paper - as it stands now, without the hints on Transformer that appear only in the supplement, the overall workings of the paper cannot be easily understood. No insight is given as to how the overall architecture solves the main motivating problem for this paper.

For their experimentation, the authors compare against a good number of competing methods. However, three of them - DCRNN, GeoMAN, and ASTGCN - use important elements of the authors' own design, namely attention-based  models and encoder-decoder architectures (GeoMan uses both). However, the authors fail to differentiate their design from these approaches.

Overall, the machinery is rather complex, underexplained, and undermotivated. The paper has major omissions and other serious presentational issues that make it very difficult to follow. The authors do not take care to point out which parts of their design are original and which are borrowed - it took much sleuthing to determine that the MGAAT differs from the GAT only in its introduction of a masking factor. As someone not previously familiar with GATs and atrous graph attention (as I suspect most of the audience would be), I found the paper very difficult going. A total overhaul of the paper would be needed in order to properly explain and motivate this work.

Overall, in its current state (not least due to presentational issues) the paper appears to be significantly below the acceptance threshold.


**Experience Assessment:**

I do not know much about this area.

**Review Assessment: Checking Correctness Of Derivations And Theory:**

I assessed the sensibility of the derivations and theory.

**Review Assessment: Checking Correctness Of Experiments:**

I assessed the sensibility of the experiments.

**Review Assessment: Thoroughness In Paper Reading:**

I read the paper at least twice and used my best judgement in assessing the paper.

---

> ### Author Response · Authors · 2019-11-15
> **We appreciate many insightful comments from this reviewer.**
>
> 1.	“No insight is given as to how the overall architecture solves the main motivating problem for this paper.” We will elaborate more on how the overall architecture solves the motivating problem。 Please see Section 1 and Section 2 for details。
>
> 2.	“ - DCRNN, GeoMAN, and ASTGCN - use important elements of the authors' own design, namely attention-based models and encoder-decoder architectures (GeoMan uses both). However, the authors fail to differentiate their design from these approaches.” -Our work significantly differs from the three methods. First, on spatial mode, none of them use any clustering techniques, while efficiently handling the unsmoothness issues. Second, on temporal mode, we use attention for temporal feature extraction, as well as encoder-decoder connection for decoding encoder outputs. In contrast, GeoMAN only has encoder-decoder attention as well as LSTM as the temporal learner. Moreover, ASTGCN does not incorporate position encoding. Third, we have included more detailed comparisons in Section 4.1.
>
> 3.	“... MGAAT differs from the GAT only in its introduction of a masking factor”
> We have improved our organization for better illustration. MGAAT is the application-level adaptation for GATs to incorporate multiple graphs and atrous attention (the masking factor). The technical details are written in section 2.7. We have removed this terminology from the novel technical part for simplicity.
>
> 4.	“… major omissions and other serious presentational issues….” –We will reorganize the whole paper and improve the presentation by following your helpful comments. Please see section 2 for details.

---

### Public Comment · ~John_Berkeley1 · 2019-11-01
**This paper does not meet the standard of ICLR**

This paper does not meet the standard of ICLR due to the following reasons:

1. Very incremental novelty
2. Major technical flaws
3. Neglect important baselines
4. Incomplete results that cannot support the validity and effectiveness of the proposed techniques
5. Impossible to reproduce and verify the results


Detailed reasons:

1. Very incremental novelty
This paper has two components — the clustered attention block and the multi-view position encoding.

The clustered attention block uses graph attention networks to model spatial dependency. However, many papers have done that before, even on the same problem. For example,

Ying Xu and Dongsheng Li, Incorporating Graph Attention and Recurrent Architectures for City-Wide Taxi Demand Prediction, International Journal of Geo-Information, 2019.

In addition, this component is very similar to the multi-head graph attention networks [Velickovic et al., ICLR’18] (https://arxiv.org/abs/1710.10903 ) and atrous attention proposed in the sparse Transformer paper [Child et al., 2019] (https://arxiv.org/abs/1904.10509 ). The only major difference is to learn the clustering assignment, which has serious flaws, as detailed later.

The multi-view position encoding is for helping Transformer to get time order information for the input time series. Transformer has been applied to forecasting time series before. In addition, the proposed multi-view position encoding is very similar to the sine/cosine encoding in the original Transformer paper with only minor differences.

Overall, this paper just simply stacks/groups several well-known techniques together without significant modifications. I think this paper is very hard to meet the bar of novelty for ICLR, as ICLR is for papers with original innovation.


2. Major technical flaws
For the most important contribution of the paper — clustering attention blocks, the paper uses the sum of “consistency”, “purity”, and “diversity” losses for clustering vertices (each vertex corresponds to a location). See Section 2.6. However, these losses are wrong.

First, the “consistency” loss encourages a location to stay in the same cluster across time. However, why should a location always stay in the same cluster? A major bus station may be more related to business regions during daytime and more related to recreation regions during nights — it should stay in different clusters at different times.

Second, the “purity” loss encourages a location to stay in just a single cluster. However, why a location cannot belong to multiple clusters if it correlates with multiple regions?

Third, the “diversity” loss encourages each cluster to be assigned with a similar number of locations. But if different regions have very different numbers of locations, should we still assign each cluster a similar number of locations?


3. Neglect important baselines
First, as this paper used Transformer to capture temporal dependency, why don’t you quantitively compare with the original Transformer [Vaswani et al., NIPS’17] (https://arxiv.org/abs/1706.03762 ) to demonstrate the benefits of the proposed techniques? The Transformer applies to the traffic forecasting problem.

Second, the author did not compare with some important state-of-the-art work in traffic forecasting [Wang et al., 2019] (https://arxiv.org/abs/1812.02019 ).


4. Incomplete results that cannot support the validity and effectiveness of the proposed techniques
First, if we ablate the clustering component (i.e., just use Transformer with MVPE), the proposed method achieves an RMSE of 9.87 which is similar to the RMSE of prior RNN/CNN-based methods. Does it mean the Transformer/MVPE component does not get any benefit over RNN/CNN-based approaches? If so, the authors should not claim something not working (i.e., Transformer/MVPE) as a contribution of your paper.

Second, the paper did not show the results on how the locations are clustered and whether they are clustered correctly. As the clustering component is the key contribution of this paper, without seeing this important result, we cannot trust the effectiveness of the approach proposed in the paper.

Third, for the multi-step prediction, the paper did not include the results for City B. Also, the paper did not mention anything about City B (e.g., the number of vertices and edges in City B). Why?


5. Impossible to reproduce and verify the results
This paper uses a private dataset and has released only part of the code (note there is public dataset available with similar scales for the problem in the paper, and there are many ways to release the dataset without breaking the double blind requirement).

---

> ### Author Response · Authors · 2019-11-15
> **We appreciate many insightful comments from “John Berkeley", even though we respectively disagree with your major comments as detailed below.**
>
> 1. “Ying Xu and Dongsheng Li (2019)… “
>
> We appreciate several references pointed out this reviewer. Although these references may consider a similar urban computing problem, our problem of interest is much more complex than them in terms of data size and regional scale.  We have only applied the  two methods  in [1] [2] to demand prediction problem for city A due to time limit and obtained the following results. It is clear that our method can achieve more than 10% improvement than those baselines.
>
> For [1], the rmse and mape are, respectively, given by 9.54 and 0.224.
> For [2], the dynamic graph model does not converge, even though we have tried very hard to tune its parameters. It seems that the method in [2] is not applicable to our problem. Thus, we use a fixed multi-graph with the rmse/mape being 9.72/0.226.
>
> [1] Incorporating Graph Attention and Recurrent Architectures for City-Wide Taxi Demand Prediction
> [2] Dynamic Spatio-temporal Graph-based CNNs for Traffic Prediction
>
> 2. “simply stacks/groups several well-known techniques together without significant modifications”. We respectively disagree with your comments. Almost all papers borrow some ideas from prior researches, the key point is whether there is any existing method that can handle our problem of interest as good as ours. As shown in our numerical studies, our method can achieve at least 10% improvement compared with those baselines including those in the references pointed by you.
>
>
> 3. “why should a location always stay in the same cluster?” In our methodology, the cluster assignment for each region is consistent within each batch (a continuous short period of time). The assignment could be different among batches (time).
> The variation of different samples within each batch is quite small for our datasets. Specifically, each batch in our design represents a short period of time. Under this batching mechanism, the spatiotemporal patterns of different samples in each minibatch are similar to each other (Please refer to appendix B for statistics). Similarly, it is reasonable to fix the cluster assignments for regions within each batch in eq(9) (eq(7) in the latest submission) and the cluster assignments are free to be different among different batches. We have added above explanation to section 2.6.
>
> "why a location cannot belong to multiple clusters if it correlates with multiple regions?"
> First, these “multiple regions” could be assigned to the same cluster, which do not violate “purity”. Second, if assigning them to more than one cluster would increase the performance significantly, the optimizer will do so. There are minor occasions of this after model converges. Besides, without significantly deteriorate the model performance, this is a trick to reduce the computational cost.
>
> "should we still assign each cluster a similar number of locations? "
> Diversity is a necessary constraint to ensure the effectiveness of clustering. Model without diversity component will sometimes assign almost all regions to one cluster. The model performance will be bad under this situation (please refer to ablation study). Balancing the load of all clusters makes sure all submodules are well-trained. Besides, this regularizer is not violating any physical rules in urban computing.
>
> Purity and Diversity: Removing these components will cause serious problems. The model without purity component assigns most of the regions to all clusters equally. Model without diversity component will assign most of the regions to one cluster.
>
> After the model converges, there are indeed some regions violating the above rules. They survive the regularizations due to their huge contributions to reducing the prediction error. However, these are only minor occasions.
>
> 4. "Does it mean the Transformer/MVPE component does not get any benefit over RNN/CNN-based approaches?"
> You cannot draw such conclusion from current evaluation results until unifying the spatial feature extractor, as well as removing unique tricks in the baseline. We are not verifying this in our evaluation due to limited space and time.
>
> "the paper did not show the results on how the locations are clustered and whether they are clustered correctly. " We have added section 4.3 to interpret the clustering result and prove the correctness of our design. The interpretation method isn't to show the spatial map of clusters, since there is no ground truth for it. Asking for the “correctness of clusters” is a logic flaw in the comment. If such “correct cluster” is available, we will directly use it rather than learn it.
>
> "for the multi-step prediction, the paper did not include the results for City B."
> We cannot make up this experiment during limited rebuttal time.
>
> 5. Reproductivity
> We have released the code. We are in the process of releasing part of our datasets used in this paper, but we have to deal with some legal issues. Our datasets are much more complex than all open datasets in terms of size and scale.

---

### Decision · Program_Chairs · 2019-12-19

**Decision:**

Reject

**Comment:**

This paper proposes an approach to handle the problem of unsmoothness while modeling spatio-temporal urban data. However all reviewers have pointed major issues with the presentation of the work, and whether the method's complexity is justified.